# Auditing Robot Learning for Safety and Compliance during Deployment

**Homanga Bharadhwaj**
School of Computer Science, Carnegie Mellon University
`hbharadh@cs.cmu.edu`

**Abstract:** Robots of the future are going to exhibit increasingly human-like and super-human intelligence in a myriad of different tasks. They are also likely going to fail and be incompliant with human preferences in increasingly subtle ways. Towards the goal of achieving autonomous robots, the robot learning community has made rapid strides in applying machine learning techniques to train robots through data and interaction. This makes the study of how best to audit these algorithms for checking their compatibility with humans, pertinent and urgent. In this paper, we draw inspiration from the AI Safety and Alignment communities and make the case that we need to urgently consider ways in which we can best audit our robot learning algorithms to check for failure modes, and ensure that when operating autonomously, they are indeed behaving in ways that the human algorithm designers intend them to. We believe that this is a challenging problem that will require efforts from the entire robot learning community, and do not attempt to provide a concrete framework for auditing. Instead, we outline high-level guidance and a possible approach towards formulating this framework which we hope will serve as a useful starting point for thinking about auditing in the context of robot learning.

**Keywords:** Robot Learning, Auditing, Safety, Compliance, Alignment

## 1 Introduction

While recent progress in robot learning has achieved significant growth in terms of being able to perform complex manipulation, locomotion, and navigation tasks with minimal hand-designed controllers and very little expert supervision [1, 2, 3, 4], deployment-specific considerations like safety, ethics, and compliance have not received their fair share of attention. Training robots to solve tasks with machine learning algorithms has the benefit of not requiring exact specification of the desired behavior, but instead providing data of how that behavior should look like or an objective function that optimizes for the desired behavior. Reinforcement learning (RL) and imitation learning (IL) are two broad classes of algorithms for robot learning, that differ with respect to the type of supervision: whether a reward function is provided, or examples of demonstrations from an expert.

Despite offering several benefits in terms of flexibility, machine learning approaches offer either very weak or no guarantees on the type of behavior a trained model might be expected to exhibit during deployment. In particular, for deep networks, generalization bounds based on distribution shift are largely trivial and strong guarantees are limited to restricted problem settings and simple architecture designs [5, 6, 7, 8]. Due to these limitations, it is very difficult to understand how accurately would the trained models align with their intended behavior. In the broader context of general AI systems, this is often termed the *AI Safety and Alignment* problem [9, 10].

Based on the limitations above, in this paper, we make the case that since robot learning involves applying machine learning algorithms to solve control problems, we need a systematic way to audit trained models prior to deploying them in real-world applications. We motivate the necessity of auditing under two lenses: safety and compliance. Safe behavior can be defined as guarantees on the robot behavior that prevent catastrophic failures from happening to the robot and those interacting with it, including humans and inanimate objects. Since the goal of developing intelligent robots is

Blue Sky Papers, 5th Conference on Robot Learning (CoRL 2021), London, UK.

to help humans by co-existing with them, we need to determine how compliant are the robots with social norms and human preferences.

In the next sections, we motive the problems of safety and compliance by grounding them in prior work, and then describe a possible approach for developing the audit framework for robot learning based on these considerations.

## 2 Safety

In the control theory literature, there are provable safety guarantees for control algorithms, for example through Hamilton-Jacobi Reachability based methods [11, 12]. Other works have provided safety and stability guarantees for RL based control problems under structural assumptions about the environment dynamics, safety structures, or access to user demonstrations [13, 14, 15]. Some other approaches have provided safety guarantees for RL without additional assumptions, but they typically satisfy the safety constraints only at convergence or have finite but non-zero failures during training [16, 17, 18].

Although there have been some prior works in safety for robot learning, like the ones above, the issues of safety have received far less attention from the community compared to ways in which *task performance* is maximized. Most of the prior approaches study safe RL for control under a constrained optimization problem where the constraints are simple checks on failures of the agent, for example the agent falling down on the ground. As documented in [18, 17], safety and task performance are sometimes at odds with each other especially when safety is enforced in the form of constraints the agent must respect during training. Hence, it is unlikely that safe behaviors would emerge automatically while trying to optimize a task-specific reward function.

As a simple example, consider a robot learning to pick and place a cup of water on the table. The task objective is usually such that the agent is rewarded for placing the cup in the desired location, but this doesn't prevent the agent from spilling water from the cup on the table and damaging potential electronic equipments on the table. One type of desired behavior from a safety perspective would be to do something for the sake of safety: in this scenario for example, the robot might push aside the electronics *before* trying to move the cup. Such commonsense reasoning comes naturally to humans, but it is tricky to determine safety objectives and constraints that would lead to such desirable safe behaviors.

The above example is a type of behavioral safety that we would want the autonomous agents to exhibit. We believe this would require moving beyond the formulation of safety as simple constraints and fixed objective functions, to human-in-the-loop settings, where humans can continuously provide interactive feedback to the robot [19, 20].

## 3 Compliance

Generally speaking, compliance refers to adhering to a rule, policy, or specification. When we train machine learning algorithms, compliance with human preferences is an implied desiderata - we expect the trained algorithm to behave according to the objective function we specified for training, and the objective function in turn is expected to be a proxy for the preferences of the algorithm designer. There are two broad challenges: *optimizing* with respect to a specified objective function, and *designing* the objective function in the first place.

Most of the machine learning community is focused on addressing the first problem and coming up with better ways to optimize either through an improved learning algorithm, or through better neural network architectures. This is usually not an issue because for a large number of supervised learning problems, like classification with cross-entropy loss, or unsupervised learning problems, like image generation with pixel-reconstruction error, the objectives work reasonably well in achieving the desired outcome. In sequential decision problems like learning to play games via reinforcement learning for example, the reward function is usually defined by the rules of the game itself and doesn't require manual designing [21]. However, when we move away from these settings into complex real-world robot control problems, it becomes increasingly unclear how the reward function should be designed [22, 23].

For example, for a task like a robot grasping a cup of water and moving it across the table to hand it to a person, the reward function for RL is unclear. Previous works have manually created dense reward functions based on some intuitions about the different stages of the process [24]. Some works have specified the reward to be $+1$ when the task is *complete* and $0$ otherwise - the so-called

sparse reward function [25]. Other approaches have sought to use human demonstrations in order to encourage the robot to learn to match the demonstrations directly (behavior cloning based methods) or learn a possible reward function from demonstrations and perform RL with the learned reward function [26, 27, 28].

Irrespective of the approach adopted in the previous example, it is not possible to guarantee that the objective function indeed corresponds to what we as the algorithm designers want from the robot. In particular, if we specify the objective *incorrectly*, we are likely to get vastly different and potentially catastrophic outcomes from the one we desire [29, 30]. For example, the robot might learn to grasp the cup and bring it to the person but drop it instead of smoothly handing it over. In order words, we might obtain a trained robot that is not *compliant* with our preferences.

## 4   Auditing Robot Learning algorithms

In light of safety and compliance considerations, we believe it is important to develop rigorous audit frameworks for robot learning that test specific failure modes of the system before deployment. To formalize the audit problem, we would come up with specifications that we want the robot to satisfy and under different controlled variations of the environment and tasks, test whether the specifications are satisfied. The outlined approach is inspired by prior work in auditing machine learning models [31]. A key desiderata for the specification is to encode human preferences about what is safe vs. unsafe and what is compliant vs. uncompliant behavior. So, the audit framework must involve a human-in-the-loop.

Based on studies in the AI Alignment literature [32, 33], purely didactic objective functions, where the human specifies an order which the robot must optimize its behavior for, leads to unintended consequences. This is because humans are themselves not adept in distilling their preferences into a crisp statement for a perfect objective function, so the objective being specified to the robot is likely imperfect. To mitigate this, instead of the human specifying an objective offline for the robot to optimize for, learning should be an iterative process of interaction between the human and the robot.

We analyze the audit framework by decomposing it into three parts: verification, verified training, and deployment. For ease of description, we assume RL as the policy training approach.

**Verification.** Consider a pool of humans, $\mathcal{H}$ that forms the set of auditors. Let $\mathcal{T}$ denote the set of tasks we want the robot to accomplish, with each task $T_i \in \mathcal{T}$ being specified with a proxy reward function $R_i$. Given task $T_i$ specified by the auditors $\mathcal{H}$, the robot executes a sequence of actions $a_t \in \mathcal{A}$ at every time-step, in states $s_t \in \mathcal{S}$, by receiving rewards $r_t \sim R_i(s_t, a_t)$ and stops (for example by executing a STOP action) when it considers the task is complete. We will talk about training in the next subsection, here we consider that training has completed, and the robot is being verified for its behavior.

The human auditors $\mathcal{H}$ inspect temporal chunks of the robot's trajectory and mark each chunk with a ✓ or a ✗. A ✓ denotes that the chunk is both safe and compliant, whereas a ✗ denotes that the chunk is either unsafe or uncompliant. An aggregate over all the auditors in $\mathcal{H}$ can be used to determine whether each chunk is ✓ or ✗. Now, an aggregate over the chunks for the trajectory can be done to determine to what extent the entire trajectory is both safe and compliant - for example by reporting the fraction of ✓ and ✗ over the entire trajectory. The same is repeated for each task in $\mathcal{T}$.

**Verified Training.** In addition to verifying whether the specifications by human auditors are satisfied, we can also train the robots such that the specifications are likely to be satisfied in the first place. In this case, the training algorithm involves the human auditors $\mathcal{H}$ in the loop. Let us assume each temporal chunk to be $\Delta_t$ such that after every $\Delta_t$ interval, the human auditors aggregate a ✓ or ✗ for that part of the robot's trajectory. For simplicity, let $\mathcal{T}$ just consists of one task $T$ informed by a proxy reward function $R$ specified initially.

The human auditors influence the training process of the robot by altering the reward function. We assume that the auditors have a bag of proxy-reward functions $\{\hat{R}_j\}$ that can be swapped with the current reward function that the robot is operating with. If the auditors determine a temporal chunk $\Delta_t$ of the robot's trajectory $\{(s_t, a_t, r_t)\}_{t=tx}^{tx+\Delta_t}$ to be either unsafe or uncompliant, then the reward function $R_i$ is swapped with $\hat{R} \in \{\hat{R}_j\}$ and the process is repeated for each chunk $\Delta_t$. In order to avoid problems with the policy being improperly optimized due to a changing reward function, $\Delta_t$ could be a *sufficiently long* interval of time.

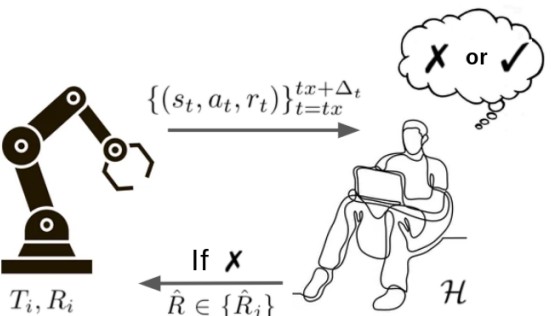

Specification sheet

| Tasks | $\dfrac{\checkmark}{\checkmark + \text{✗}}$ |
|---|---|
| Task 1 | $0.87 \pm 0.11$ |
| Task 2 | $0.52 \pm 0.08$ |
| Task 3 | $0.67 \pm 0.12$ |

Figure 1: Illustration of the **audit framework** with *verified training* and *deployment* phases. The robot is initially assigned a proxy reward function $R_i$ to optimize for solving task $T_i$. The human auditors $\mathcal{H}$ observe a temporal chunk $\Delta_t$ of the robot's trajectory $\{(s_t, a_t, r_t)\}_{t=tx}^{tx+\Delta_t}$ and determine whether it is either unsafe/uncompliant (✗) or not (✓). If it is marked ✗, then the reward function $R_i$ is swapped with $\hat{R} \in \{\hat{R}_j\}$ and the process is repeated for each chunk $\Delta_t$. After this verified training phase, in the final *verification* phase, the entire trajectory of the robot for each task is marked with ✓or ✗for each chunk, and the aggregate fraction is reported in a specification sheet. The S.D. is over differ humans in the set $\mathcal{H}$. When deploying the robot, it is accompanied by the specification sheet, so that the end-user can determine in which tasks to use the robot and where to not use it.

An additional desiderata in this setup could be that the robot maintains a distribution over the parameters of the reward function $R$ to take into account that $R$ would change during training through human-intervention, and optimizes an ensemble of policies by sampling different reward functions from this distribution. When $R$ changes to some $\hat{R}$, the distribution would be maintained over this new reward function.

**Deployment.** After verified training, followed by a final verification step for all tasks, the auditors would make a specification sheet that lists each task $T_i \in \mathcal{T}$ and the fraction of ✓and ✗for them $\frac{\checkmark}{\checkmark + \text{✗}}$. When the robot is finally deployed, it would be accompanied by this specification sheet to help the end-user determine, in which tasks to use the robot and where to avoid using it.

## 5    Discussion

In this paper, we proposed the problem of auditing robot learning algorithms under the lens of safety and compliance and provided high-level ideas about a possible approach to designing an audit framework. Through this paper, our main objective is to communicate the importance and immediate relevance of problems studied in the *AI safety* and *AI alignment* communities, to robot learning. Since the robots we are developing are becoming increasingly intelligent and autonomous, we must devise formal approaches to audit these robots for ensuring they are safe to interact with and their behaviors are compatible with human preferences.

We would like to emphasize that the description in section 3 as is would not lead to a *practical* framework, for a number of reasons, and these open up a lot of avenues for research in this direction. First and foremost, it is impractical to assume that humans would be available to babysit the robot during training, for hours and in some cases even days, while intervening at appropriate times with a modified reward function. So it is imperative to come up with a scheme, where the intervention frequency with humans is minimized and parts of the framework are more automated. Second, a changing reward function is likely to present optimization challenges for policy learning with RL. To mitigate this and alleviate issues with catastrophic forgetting of neural network policies, works in continual learning could be useful.

Third and most importantly, the set of human auditors need to come up with a reasonable set of reward functions that can be used to replace the current reward function that the robot is optimizing the policy with. These reward functions which for example could be differently parameterized or have completely different functional forms from each other, reflect the *imperfect* distillation of the human's preferences. So, it is important to make the switch in reward functions when the behavior observed is undesirable, instead of letting the robot operate with a fixed offline reward function.

We believe that our paper and the ideas we presented would lead to increased synergy between the AI safety/alignment and robot learning communities, and serve as a starting point for formal frameworks in auditing robot learning.

## Acknowledgements

I thank Abhinav Gupta, Shubham Tulsiani, Victoria Dean, Vikash Kumar, Liyiming Ke, Aravind Rajeswaran, Pedro Morgado, Animesh Garg, De-An Huang, Chaowei Xiao, Anima Anandkumar, Florian Shkurti, Samarth Sinha, Sergey Levine, Dumitru Erhan, Gaoyue Zhou, Keene Chin, Yufei Ye, Shikhar Bahl, Sudeep Dasari, Raunaq Bhirangi, Helen Jiang, Jason Zhang, Himangi Mittal, Jianren Wang, and Sally Chen for helpful discussions and inspiration.

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
