# OpenReview forum: "Auditing Robot Learning for Safety and Compliance during Deployment"
_robot-learning.org/CoRL/2021/Conference/Blue_Sky — CoRL 2021, Blue Sky_

### Official Review · Reviewer_zcpS · 2021-07-31

**Novelty:** Fair
**Impact:** 2
**Clarity Of Presentation:** Good

**Recommendation:**

Weak Reject: I recommend rejecting the paper, but will not argue for my recommendation if the majority of other reviewers have a different opinion.

**Summary:**

This paper highlights the problem of auditing robot behaviors (learned via reinforcement learning) for safety and compliance. Safe behavior is defined as behavior that prevents catastrophic failures to the robot and agents around it. Compliant behavior is defined in the spirit of the value alignment problem (i.e., complying with humans’ preferences). The paper does not provide concrete proposals on how to address these challenges. Instead, the goal is to outline high-level guidance on formulating a framework for robot auditing.

**Summary Of Recommendation:**

Overall, I do not believe that this paper makes a significant or novel contribution to the discussion on safety in robot learning. The primary points made in the paper are uncontroversial and already largely appreciated by the community. In particular, the paper highlights two challenges with learning-based control of robots: (i) safety, and (ii) compliance. There is currently significant activity within the robotics community on ensuring safety of learning-based systems. The paper cites work that seeks to ensure safety during the training process, e.g., references [13]-[18]. There is also work on guaranteeing safety during deployment in novel environments (i.e., safe generalization):
- A. Majumdar, A. Farid, and A. Sonar, “PAC-Bayes Control: Learning Policies that Provably Generalize to Novel Environments”,
- A.Z. Ren and A. Majumdar, “Generalization Guarantees for Imitation Learning”,

along with work on performing novelty detection (i.e., out-of-distribution detection) in RL contexts; see the following papers (and many others):

- F. Cai and X. Koutsoukos, “Real-time out-of-distribution detection in learning-enabled cyberphysical systems”
- I. Greenberg and S. Mannor, “Detecting rewards deterioration in episodic reinforcement learning”.

There is also a growing body of work on ensuring compliance in the literature on the value alignment problem (a number of papers are cited in the paper).

Given that the paper highlights problems that are already largely appreciated by the community, the burden is then to either provide a new perspective on these problems or propose concrete proposals for tackling them. The paper does not do this; instead, the paper outlines a very high-level framework which includes verification, verified training, and auditing during deployment. Again, there is a body of work in each of these areas (e.g., in the literatures on adversarial training, verification and validation).

Finally, the paper takes a relatively narrow perspective on the problems of safety and compliance in RL. In particular, the paper does not discuss much work beyond the academic research community. The problem of safety of learning-enabled systems is a major concern for regulatory agencies (e.g., the Federal Aviation Administration in the US). Such agencies are actively exploring possible solutions to these problems; see, for example, a recent report from the G-34 committee of the Society of Automative Engineers (SAE):

SAE G-34 committee, “AIR6988 - Artificial Intelligence in Aeronautical Systems: Statement of Concerns”.

Overall, given that this paper neither highlights novel concerns nor proposes concrete directions, it is difficult to see how it will help advance the state of the art in this area.

---

### Official Review · Reviewer_bXz3 · 2021-08-25

**Novelty:** Very Good
**Impact:** 3
**Clarity Of Presentation:** Very Good

**Recommendation:**

Weak Accept: I recommend accepting the paper, but will not argue for my recommendation if the majority of other reviewers have a different opinion.

**Summary:**

This work addressed the practical considerations of safety and compliance for the deployment of robot learning systems. The key idea is to have humans systematically audit a trained model prior to using it in real-world applications. The authors examined prior work in the robot learning literature on safety guarantees and compliance with human preferences. It proposed a framework for a group of human auditors to inspect a robot's execution and supply feedback to chunks of robot trajectories. The feedback is in the form of checkmarks and crosses. This auditing process will produce a specification sheet that informs the end-user of the limitations and risks of robot behaviors.

**Summary Of Recommendation:**

This paper tackled a crucial problem of safety and compliance with a new angle of model auditing. The authors motivated the problem of safety and compliance well and examined related work in both aspects. The proposed algorithm in Sec. 4 seems to be a natural design of an initial audit framework. Though several important design choices need careful thinking:

1) the human cost involved in the auditing process and how to deal with disagreement and inconsistency;
2) how to make sure the trajectories are sufficiently discriminative for soliciting human feedback and how to handle delayed effects (an action seemingly benign could cause a catastrophe later in the future);
3) how to use the specification sheet to guide real-world deployment, and further, how to use this specification to improve the robot's behaviors.

The authors are upfront that the description in sec 3 would not lead to a practical framework. Nonetheless, the reviewer believed that this manuscript, as a blue-sky paper, has pinpointed an important research problem, which has great potential to unlock fruitful future research. For this reason, the reviewer is leaning towards accepting this paper.

---

### Decision · Program_Chairs · 2021-10-01

**Decision:**

Accept

**Comment:**

The paper argues for a human audit of trained models for safety and compliance. While the method might not be directly practical, it offers a unique perspective in a not studied-enough domain, and might spark a good discussion in the field.